# MATT: Random Local Implicit Purification for Defending Query-based Attacks

## Abstract

Black-box query-based attacks constitute significant threats to Machine Learning as a Service (MLaaS) systems since they can generate adversarial examples without accessing the target model's architecture and parameters. Traditional defense mechanisms, such as adversarial training, gradient masking, and input transformations, either impose substantial computational costs or compromise the test accuracy of non-adversarial inputs. To address these challenges, we propose an efficient defense mechanism, MATT, that employs random patch-wise purifications with an ensemble of lightweight purification models. These models leverage the local implicit function and rebuild the natural image manifold with low inference latency. Our theoretical analysis suggests that this approach slows down the convergence of query-based attacks while preserving the average robustness improvement by combining randomness and purifications. Extensive experiments on CIFAR-10 and ImageNet validate the effectiveness of our proposed purifier-based defense mechanism, demonstrating significant improvements in classifier robustness against query-based attacks.

## 1 Introduction

Deep neural networks (*DNNs*), while presenting remarkable performance across various applications, are mostly leaning to become subject to *adversarial attacks*, where even slight perturbations to the inputs can severely compromise their predictions (Szegedy et al., 2014). This notorious vulnerability significantly challenges the inherent robustness of DNNs and could even make the situation much worse when it comes to security-critical scenarios, such as facial recognition (Dong et al., 2019) and autonomous driving (Cao et al., 2019). Accordingly, attackers have devised both *white-box attacks* if having full access to the DNN model and *black-box attacks* in case the model is inaccessible. While black-box attacks appear to be more challenging, it is often considered a more realistic threat model, and its state-of-the-art (SOTA) could leverage a limited number of queries to achieve high successful rates against closed-source commercial platforms, i.e., Clarifai (Clarifai, 2022) and Google Cloud Vision API (Google, 2022), presenting a disconcerting situation.

Defending black-box query-based attacks in real-world large-scale Machine-Learning-as-a-Service (*MLaaS*) systems calls for an extremely low extra inference cost. This is because business companies, such as Facebook (VentureBeat, 2022), handle millions of image queries daily and thereby increase the extra cost for defense a million-fold. This issue prohibits testing time defenses to run multiple inferences to achieve *certified robustness* (Cohen et al., 2019; Salman et al., 2020b). Moreover, training time defenses, i.e., retraining the DNNs with large datasets to enhance their robustness against adversarial examples (e.g., *adversarial training* (Madry et al., 2018) and *gradient masking* (Tramèr et al., 2018)), impose substantial economic and computational costs attributed to the heavy training expense. Therefore, there is a critical need for a lightweight yet effective strategy to perform adversarial purifications to enable one inference cost for achieving robustness.

Given the aforementioned challenges, recent research efforts have been devoted to either eliminating or disturbing adversarial perturbations prior to the forwarding of the query image to the classifier. Nevertheless, the existing methods that include both heuristic transformations and neural network-based adversarial purification models have certain limitations in removing adversarial perturbations. While heuristic transformation methods cause minimal impact on cost, they merely disrupt adversarial perturbations and often negatively impact the testing accuracy of non-adversarial inputs Xu

et al. (2018); Qin et al. (2021). Moreover, purification models aiming to completely eradicate adversarial perturbations can even exceed the computational burden of the classifier itself Carlini et al. (2023). Consequently, there have been no effective defense mechanisms that can achieve both high robustness and low computational cost against query-based attacks.

In this paper, we propose a novel random patch-wise image purification mechanism leveraging the local implicit function to improve the robustness of the classifier against query-based attacks. The idea of local implicit function is first proposed for super-resolution tasks (Lim et al., 2017; Zhang et al., 2018), and has recently been showing potentiality in defending against white-box attacks with low computational cost (Ho & Vasconcelos, 2022). Nonetheless, we find that the naive local implicit function combined with the classifier forms a new black-box system that is still vulnerable to query-based attacks (**6.8%** robust accuracy on ImageNet datasets under strong attack), and our theoretical analysis attributes this to the lack of randomness inside the purifier. Although randomness can be introduced using an ensemble of purifiers, the inference cost of encoding-querying structure within the local implicit function almost increases linearly with the number of purifiers. To address these challenges, we design an end-to-end purification model and only approximate the local implicit function in a local patch using a randomly chosen purifier from a diversified pool. Our method allows a significant diversity gain with more purifiers while keeping almost the same inference cost. Our theoretical analysis shows that our system is more robust with more different purifiers and slows down the convergence of the query-based attacks.

Our contributions are summarized as follows:

- We propose a novel defense mechanism using the local implicit function to randomly purify patches of the image to improve the robustness of the classifier. Our work is the first to extend the local implicit function to defend against query-based attacks.

- We provide a theoretical analysis on the effectiveness of our proposed purifier-based defense mechanism based on the convergence of black-box attacks. Our theoretical analysis points out the potential vulnerabilities of deterministic transformation functions and suggests the robustness of our system increase with the number of purifiers.

- Our theoretical investigation reveals the connection between the attack's convergence rate and transformation function used under the black-box setting, offering a new perspective on understanding the efficacy of defense mechanisms employed at the preprocessing stage.

- We conduct extensive experiments on CIFAR-10 and ImageNet on current SOTA query-based attacks and verify the effectiveness of our methods in defending query-based attacks.

## 2 RELATED WORK

**Query-based Attacks.** Query-based attacks, continually querying the models to generate adversarial examples, are categorized as either *score-based* or *decision-based*, based on their access to confidence scores or labels, respectively.

Score-based attacks perceive the MLaaS model, inclusive of pre-processors, the core model, and post-processors, as a black-box system. The objective function, in this case, is the marginal loss of the confidence scores, as depicted in Equation (1). Black-box optimization techniques, such as *gradient estimation* and *random search*, can be harnessed to tackle this issue. Ilyas et al. (2018) developed the pioneering limited query score-based attack using Natural Evolutionary Strategies (NES) for gradient estimation. This sparked a flurry of subsequent studies focusing on gradient estimation, including ZO-SGD (Liu et al., 2019) and SignHunter (Al-Dujaili & O'Reilly, 2020). The current cutting-edge score-based attack, Square attack (Andriushchenko et al., 2020), utilizes random search through localized patch updates. It is frequently cited as a critical benchmark in evaluating model robustness (Croce et al., 2021). Other attack algorithms like SimBA (Guo et al., 2019) also employ random search but not as effectively as the Square attack.

Regarding decision-based attacks, the label information typically serves as a substitute since confidence values are not provided. An early work by Ilyas et al. (2018) uses NES to optimize a heuristic proxy with limited queries. The gradient estimation method for decision-based attacks evolves to be more efficient by forming new optimization-based problems (e.g., OPT (Cheng et al., 2019)) and utilizing the sign of the gradient instead of the estimated value (e.g., Sign-OPT (Cheng et al., 2020)

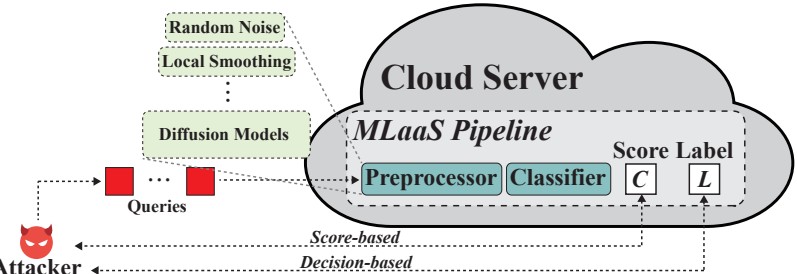

Figure 1: Illustration of the MLaaS system with defense mechanism within preprocessor under attack. The attackers can query the model with input $x$ and get the returned information $\mathcal{M}(x)$ which can be the confidence scores or the predicted label.

and HopSkipJump (Chen et al., 2020)). While direct search used in Boundary Attack (Brendel et al., 2018) is the first decision-based attack, the HopSkipJump Attack is currently considered the most advanced attack.

**Adversarial Purification.** Recently, the employment of testing-time defenses has witnessed a significant surge, primarily for the purpose of adversarial purification in order to improve the model's robustness. Yoon et al. (2021) leverages a score-based generative model to eliminate the adversarial perturbations. Techniques in self-supervised learning like contrastive loss are used by Mao et al. (2021) to purify the image. Subsequent to the success attained by diffusion models, they have been deployed for the development of certified robustness for image classifiers (Nie et al., 2022; Carlini et al., 2023). Nonetheless, due to the vast number of parameters contained within diffusion models, they suffer from much lower inference speed compared to classifiers. Recently, the introduction of the local implicit function model for defending white-box attacks has been noted (Ho & Vasconcelos, 2022). However, they only apply the purifier model trained on a handful of white-box attacks, and haven't established a resilient defense system with any theoretical assurance for defending black-box attacks. In our paper, we have redesigned the network structure by eliminating multi-resolution support, resulting in inference time acceleration by a factor of four. Moreover, the design of our defense mechanism ensure the inference speed does not increase linearly with the number of purifier models, which is the case for DISCO (Ho & Vasconcelos, 2022) when randomness is introduced. Furthermore, we offer a theoretical analysis emphasizing the efficacy of our proposed purifier-based defense mechanism against query-based attacks, centering around the convergence of black-box attacks. We provide more detailed background information on other general defense mechanism for the readers interest in Appendix A.

## 3 PRELIMINARIES

### 3.1 THREAT MODEL

In the context of black-box query-based attacks, our threat model presumes that attackers possess only a limited understanding of the target model. Their interaction with the model, which is typically hosted on cloud servers, is restricted to querying the model and receiving the resultant data in the form of confidence scores or labels. They lack additional insight into the model or the datasets used. An illustration of the MLaaS system under attack is shown in Figure 1.

### 3.2 QUERY-BASED ATTACKS

#### 3.2.1 SCORE-BASED ATTACKS

Assume a classifier $\mathcal{M} : \mathcal{X} \rightarrow \mathcal{Y}$ is hosted on a cloud server, where $\mathcal{X}$ is the input space and $\mathcal{Y}$ is the output space. Attackers can query this model with an input $x \in \mathcal{X}$ and obtain the corresponding output $\mathcal{M}(x) \in \mathcal{Y}$. In scenarios where the model's output, frequently in the form of a confidence score, is directly returned to the attackers, this is identified as a score-based attack.

Table 1: List of heuristic transformations and SOTA purification models. Randomness is introduced in DISCO (Ho & Vasconcelos, 2022) by using an ensemble of DISCO models to generate features for random coordinate querying, which is of high computational cost.

| Method | Randomness | Type | Inference Cost |
|---|---|---|---|
| Bit Reduction (Xu et al., 2018) | ✗ | Heuristic | Low |
| Local Smoothing (Xu et al., 2018) | ✗ | Heuristic | Low |
| JPEG Compression (Raff et al., 2019) | ✗ | Heuristic | Low |
| Random Noise (Qin et al., 2021) | ✓ | Heuristic | Low |
| Score-based Model (Yoon et al., 2021) | ✓ | Neural | High |
| DDPM (Nie et al., 2022) | ✓ | Neural | High |
| DISCO (Ho & Vasconcelos, 2022) | ✗ / ✓ | Neural | Median / High |
| MATT (Ours) | ✓ | Neural | Median |

In this setting, attackers generate an adversarial example $x_{adv}$ based on the clean example $x$ with the true label $y$, aiming to solve the following optimization problem to execute an untargeted attack:

$$\min_{x_{adv} \in \mathcal{N}_R(x)} f(x_{adv}) = \min_{x_{adv} \in \mathcal{N}_R(x)} (\mathcal{M}_y(x_{adv}) - \max_{j \neq y} \mathcal{M}_j(x_{adv})). \tag{1}$$

Here, $\mathcal{N}_R(x) = \{x' | \|x' - x\|_p \leq R\}$ represents a $\ell_p$ ball around the original example $x$. In the case of targeted attacks, $j$ is fixed to be the target label instead of the index of the highest confidence score excluding the true label. The attack is deemed successful if the value of the objective function is less than zero.

While projected gradient descent algorithm is used in white-box attacks, the attackers under black-box setting do not have access to the gradient information. Thus, black-box algorithms usually leverage the following techniques to estimate the function descent direction: *gradient estimation* and *heuristic search*. Further details of these techniques are included in Appendix B.

### 3.2.2 DECISION-BASED ATTACKS

Attackers have explored various ways of forming optimization problems for decision-based attacks, since the landscape of the objective function is discontinuous Cheng et al. (2019). For example, Ilyas et al. (2018) uses a proxy of the objective function, Cheng et al. (2020) and Aithal & Li (2022) forms new problems based on geometry, and Chen et al. (2020) deal with the original problem but with the sign of the gradient. Our theoretical analysis can be also applied to decision-based attacks as they employ similar techniques in solving this black-box optimization problem.

### 3.3 ADVERSARIAL PURIFICATION

Adversarial purification has recently emerged as a central wave of defense against adversarial attacks, which aims to remove or disturb the adversarial perturbations via *heuristic transformations* and *purification models*. We have provided a list of widely heuristic transformations and SOTA purification models in Table 1.

**Heuristic Transformations.** Heuristic transformations are unaware of the adversarial perturbations and aim to disturb the adversarial perturbations by shrinking the image space (Bit Reduction and Local Smoothing *etc.*) or deviating the gradient estimation (Random Noise).

**Purification Models.** Powerful purification models are trained to remove the adversarial perturbations and project the adversarial images back to the natural image manifold. Popular purification models include Score-based Model (Yoon et al., 2021) and DDPM (Nie et al., 2022), and local implicit purification models, such as DISCO (Ho & Vasconcelos, 2022). Among them, only the local implicit purifier has moderate inference cost and be suitable for defending query-based attacks.

With defense mechanisms deployed as pre-processors in the MLaaS system as shown in Figure 1, the attackers need to break the whole MLaaS pipeline to achieve a successful attack. While randomness is considered as a key factor of improving the robustness of such systems (Raff et al., 2019; Sitawarin et al., 2022), naively introducing them by ensembling multiple purifiers (DISCO) will lead to a linear increase in the inference cost.

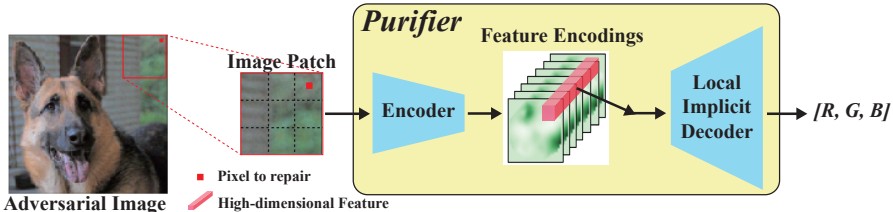

Figure 2: An illustration of repairing a pixel with our end-to-end purification model. The encoder diffuse nearby information of the pixel into its high-dimensional feature. Then the decoder reconstruct its RGB value with respect to this feature information. Note that the inference of pixels of one image patch can be performed in parallel in a batch.

## 4 RANDOM LOCAL IMPLICIT PURIFICATION

### 4.1 OUR MOTIVATION

While purifiers can execute adversarial purification on incoming images, our research, as elucidated in section 4.3 and substantiated in section 5.2, suggests that *a single deterministic purifier* cannot enhance the system's robustness. A straightforward ensembling method, albeit theoretically sound, increases the inference cost linearly with the number of purifiers, rendering it nonviable for real-world MLaaS systems. We address this issue by developing an end-to-end purification model applying local implicit function to process input images of any dimension. We further introduce a novel random patch-wise purification algorithm that capitalizes on a group of purifiers to counter query-based attacks. Our theoretical findings illustrate that the augmented robustness of our system is directly proportional to the number of the purifiers. Importantly, this approach maintains a fixed inference cost, regardless of the number of purifiers, aligning well with practical real-world MLaaS systems (refer to Appendix C for details).

### 4.2 IMAGE PURIFICATION VIA LOCAL IMPLICIT FUNCTION

Under the hypothesis that natural images lie on a low-dimensional manifold in the high-dimensional image space, adversarial perturbations can be viewed as a deviation from the natural manifold. Assume that we have a purification model $m(x) : \mathcal{X} \to \mathcal{X}$ that can project the adversarial images back to the natural manifold. If the attackers are generating adversarial examples $x'$ from the original images $x$ randomly drawn from the natural image manifold distribution $\mathcal{D}$, the purification model $m(x)$ can be trained to minimize the following loss:

$$\mathcal{L} = \mathbb{E}_{\mathcal{D}} \|x - m(x')\| + \lambda \mathbb{E}_{\mathcal{D}} \|x - m(x)\|, \tag{2}$$

where $\lambda$ controls the trade-off between the two terms. A larger $\lambda$ means a lower deviation from clean images. In practice, the second term is often ignored.

**Efficient Unique Design.** Based on prior works on local implicit function, we design an end-to-end purification model that can be trained with the above loss function, shown in Figure 2. Different from the first attempt of using an implicit function for defending white-box attack (Ho & Vasconcelos, 2022), we remove the multi-scale support by eliminating the positional encoding (structure level) and local ensemble inference (implementation level). By doing so, we achieve a 4x inference time speedup. The detailed introduction of this speed up can be found in Appendix F.1.

**Random Patch-wise Purification.** Aside from the unique design of the purification model, our main contribution lies in random patch-wise reconstruction. Although the purification model can take in images of any size and reconstruct a local area of the image, the former practice reconstructs the pixel by encoding the whole image and perform random selection from the output features to introduce randomness, as shown in Figure 3. However, since the encoder is the bottleneck of the whole model, this practice leads to a almost linear increase in the inference time with the number of purifiers. In contrast, we propose to reconstruct the pixel only using nearby pixels and introduce randomness by using a pool of purifiers. We have validated this inference speedup in Appendix C.

The comparison of encoding process from the previous method and our method is shown in Figure 3. Although each purifier can be deterministic, the randomness is introduced by randomly picking

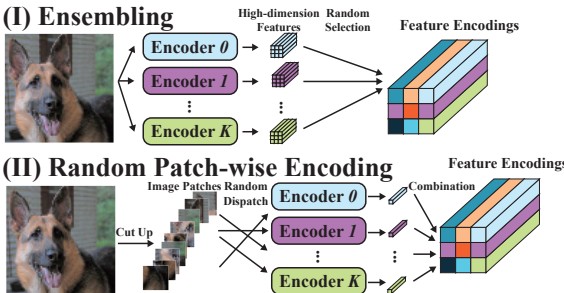

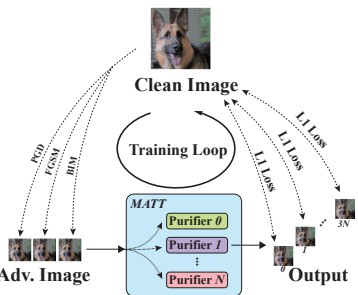

Figure 3: An illustration of the encoding process of ensembling (Ho & Vasconcelos, 2022) and our method. **Ensembling:** Ensembling method first encodes the image into multiple high-dimension features and then randomly combines them to form the final feature representation. **Random Patch-wise:** Our method split the images into patches, forward them to randomly selected encoders, and use the combination as the final feature.

Figure 4: The training process of MATT. Firstly, different adversarial images are generated using various white-box attack algorithms. Then, we use every purifier in MATT to perform the purification on the adversarial images. Thus, we have $3 \times N$ training samples for each clean image (3 attack algorithms and $N$ purifiers). Finally, we train the purifiers with these samples using $\ell_1$ loss

a purifier from the pool and performing purification on randomly selected image patches. This practice enables us to largely increase the diversity of purifiers and maintain a moderate inference time. Moreover, it allows us to actually use the purifier from a larger purification function space, if we view the combinations of used purifiers as a new purifier. The full training process is shown in Figure 4. More on the training details can be found in Appendix F.

### 4.3 THEORETICAL ANALYSIS AGAINST GRADIENT-BASED ATTACKS

Assume we have $K+1$ purifiers $\{\boldsymbol{m}_0, \ldots, \boldsymbol{m}_K\}$, the output of the new black-box system containing the $i$-th purifier is defined as $F^{(i)}(\boldsymbol{x}) = f(\boldsymbol{m}_i(\boldsymbol{x}))$. Without loss of generality, we now perform analysis on breaking the system of the purifier $\boldsymbol{m}_0$, denoted as $F(\boldsymbol{x}) = f(\boldsymbol{m}_0(\boldsymbol{x}))$. Our following analysis utilizes the $\ell_2$-norm as the distance metric, which is the most commonly used norm for measuring the distance between two images.

Suppose the index of two independently drawn purifiers in our defense are $k_1$ and $k_2$, the attacker approximate the gradient of the function $F(\boldsymbol{x})$ with the following estimator:

$$G_{\mu,K} = \frac{f(\boldsymbol{m}_{k_1}(\boldsymbol{x} + \mu \mathbf{u})) - f(\boldsymbol{m}_{k_2}(\boldsymbol{x}))}{\mu} \mathbf{u}. \tag{3}$$

*The above gradient estimator provides an unbiased estimation of the gradient of the function:*

$$F_{\mu,K}(\boldsymbol{x}) = \frac{1}{K+1} \sum_{k=0}^{K} f_\mu(\boldsymbol{m}_k(\boldsymbol{x})), \tag{4}$$

where $f_\mu$ is the gaussian smoothing function of $f$. The detailed definition of the gaussian smoothing function is included in Appendix G.1. So now we can see that the convergence of the black-box attack is moving towards an averaged optimal point of the functions of the systems formed with different purifiers, which suggests an *averaged robustness* across different purifiers.

Now we assume the purifiers has the following property:

$$\|\boldsymbol{m}_i(\boldsymbol{x}) - \boldsymbol{m}_j(\boldsymbol{x})\| < \nu, \quad \forall i, j \in \{0, \ldots, K-1\} \tag{5}$$

where $\nu$ can reflect the diversity of the purifiers. We cannot directly measure $\nu$, but we intuitively associate it with the number of purifiers. **The larger the number of purifiers, the larger $\nu$ is.**

We have the following assumptions for the original function $f(\boldsymbol{x})$:

**Assumption 1.** *$f(\boldsymbol{x})$ is Lipschitz-continuous, i.e,, $|f(\boldsymbol{y}) - f(\boldsymbol{x})| \leq L_0(f)\|\boldsymbol{y} - \boldsymbol{x}\|$.*

**Assumption 2.** *$f(\boldsymbol{x})$ is continuous and differentiable, and $\nabla f(\boldsymbol{x})$ is Lipschitz-continuous, i.e.,, $|\nabla f(\boldsymbol{y}) - \nabla f(\boldsymbol{x})| \leq L_1(f)\|\boldsymbol{y} - \boldsymbol{x}\|$.*

For the purifiers, we assume each dimension of their output also has the property in Assumption 1 and Assumption 2. Then, we denote $L_0(\boldsymbol{m}) = \max_i L_0(m_i)$ and $L_1(\boldsymbol{m}) = \max_i L_1(m_i)$, where $m_i$ is the $i$-th dimension of the output of the purifier $\boldsymbol{m}$.

**Notations.** We denote the sequence of standard Gaussian noises used to approximate the gradient as $\mathbf{U}_t = \{\mathbf{u}_0, \ldots, \mathbf{u}_t\}$, with $t$ to be the update step. The purifier index sequence is denoted as $\mathbf{k}_t = \{\mathbf{k}_0, \ldots, \mathbf{k}_t\}$. The generated query sequence is denoted as $\{\boldsymbol{x}_0, \boldsymbol{x}_1, \ldots, \boldsymbol{x}_Q\}$. $d = |\mathcal{X}|$ as the input dimension.

With the above definitions and assumptions, we have Theorem 1 for the convergence of the gradient-based attacks. The detailed proof is included in Appendix G.2.

**Theorem 1.** *Under Assumption 1, for any $Q \geq 0$, consider a sequence $\{\boldsymbol{x}_t\}_{t=0}^{Q}$ generated using the update rule of gradient-based score-based attacks, with constant step size, i.e.,, $\eta = \sqrt{\frac{2R\epsilon}{(Q+1)L_0(f)^3 d^2}} \cdot \sqrt{\frac{1}{L_0(\boldsymbol{m}_0)\gamma(\boldsymbol{m}_0,\nu)}}$, with $\gamma(\boldsymbol{m}_0, \nu) = \frac{4\nu^2}{\mu^2} + \frac{4\nu}{\mu}L_0(\boldsymbol{m}_0)d^{\frac{1}{2}} + L_0(\boldsymbol{m}_0)^2 d$. Then, the squared norm of gradient is bounded by:*

$$\frac{1}{Q+1}\sum_{t=0}^{Q}\mathbb{E}_{\mathbf{U}_t,\mathbf{k}_t}[\|\nabla F_{\mu,K}(\boldsymbol{x}_t)\|^2] \leq \sqrt{\frac{2L_0(f)^5 R d^2}{(Q+1)\epsilon}} \cdot \sqrt{\gamma(\boldsymbol{m}_0,\nu)L_0(\boldsymbol{m}_0)^3} \tag{6}$$

*The lower bound for the expected number of queries to bound the expected squared norm of the gradient of function $F_{\mu,K}$ of the order $\delta$ is*

$$O(\frac{L_0(f)^5 R d^2}{\epsilon\delta^2}\gamma(\boldsymbol{m}_0,\nu)L_0(\boldsymbol{m}_0)^3) \tag{7}$$

**Single Deterministic Purifier.** Setting $\nu$ to 0, we have $\gamma(\boldsymbol{m}_0,0)L_0(\boldsymbol{m}_0)^2 = L_0(\boldsymbol{m}_0)^5$, which is the only introduced term compared to the original convergence rate (Nesterov & Spokoiny, 2017) towards $f(\boldsymbol{x})$. Meanwhile, the new convergence point becomes $F_{\mu}^*(\boldsymbol{x})$. We have the following conclusion for the convergence of the attack:

- **Influence of $L_0(\boldsymbol{m}_0)$:** For input transformations that *shrink* the image space, since their $L_0(\boldsymbol{m}_0) < 1$, they always allow a *faster* rate of convergence for the attack. For neural network purifiers, the presence of this term means their vulnerabilities is introduced into the black-box system, making it hard to quantify the robustness of the system.

- **Optimal point $F_{\mu}^*(\boldsymbol{x})$:** By using a deterministic transformations, the optimal point of the attack is changed from $f^*$ to $F_{\mu}^*(\boldsymbol{x})$. If we can inversely find an adversarial image $\boldsymbol{x}^* = \boldsymbol{m}(\boldsymbol{x}^*)$, the robustness of the system is not improved at all. *No current work can theoretically eliminate this issue.* This may open up a new direction for future research.

> **Research implications.** From the above analysis, we can see that a single deterministic purifier may *1) accelerate* the convergence of the attack, and *2) cannot protect* the adversarial point from being exploited.

**Pool of Deterministic Purifiers.** The introduced term $\gamma(\boldsymbol{m}_0,\nu)L_0(\boldsymbol{m}_0)^2$ increase quadratically with $\nu$. This along with our intuition mentioned above suggests that *the robustness of the system increases with the number of purifiers*. While adversarial optimal points persist, the presence of multiple optimal points under different purifiers serve as the *first* trial to enhance the robustness of all purification-based methods.

To validate our theoretical analysis, we first conduct experiments on a subset of the CIFAR-10 dataset (Krizhevsky, 2009) with a ResNet-18 model (Dadalto, 2022) as the classifier. The general settings are the same as used in section 5. We use the Square Attack (Andriushchenko et al., 2020) as the attack algorithm. The convergence of the attack against our model and other input transformations is shown in Figure 5. We can see a clear acceleration of the convergence of the attack with the introduction of transformations that *shrink* the image space and powerful deterministic models fails to improve the robustness of the system. Another validation of our theoretical analysis is shown in Figure 6 for proving the robustness of the system increases with the number of purifiers (associated with $\nu$).

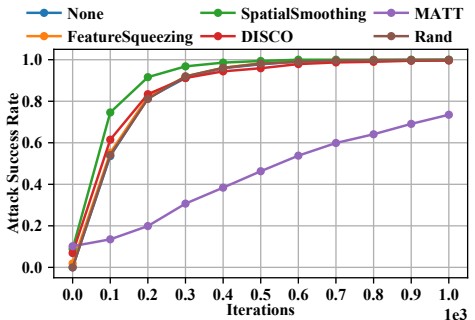 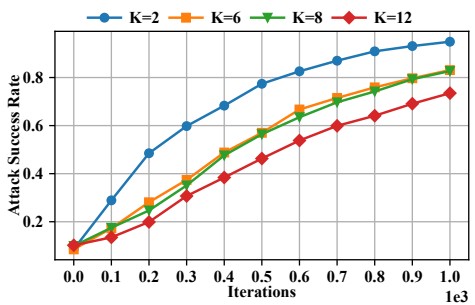

Figure 5: The convergence of the Square Attack on CIFAR-10 using different heuristic transformations and purifiers.

Figure 6: The convergence of the Square Attack on CIFAR-10 using with different numbers of purifiers used.

## 4.4 THEORETICAL ANALYSIS AGAINST GRADIENT-FREE ATTACKS

The heuristic direction of random search becomes:

$$H_K(\boldsymbol{x}) = f(\boldsymbol{m}_{k_1}(\boldsymbol{x} + \mu\mathrm{u})) - f(\boldsymbol{m}_{k_2}(\boldsymbol{x} + \mu\mathrm{u})). \tag{8}$$

**Theorem 2.** *Under Assumption 1, using the update in Equation (8),*

$$P(Sign(H(\boldsymbol{x})) \neq Sign(H_K(\boldsymbol{x}))) \leq \frac{2\nu L_0(f)}{|H(x)|} \tag{9}$$

A similar increase in the robustness as Theorem 1 can be observed with the increase of $\nu$. The detailed proof is included in Appendix G.3. This ensures the robustness of our defense against gradient-free attacks.

## 5 EVALUATION

### 5.1 EXPERIMENT SETTINGS

**Datasets and Classification Models.** For a comprehensive evaluation of MATT, we employ two widely used benchmark datasets for testing adversarial attacks: CIFAR-10 (Krizhevsky, 2009) and ImageNet (Deng et al., 2009). Our evaluation is conducted on two balanced subsets, which contain 1,000 and 2,000 *correctly classified* test images from CIFAR-10 and ImageNet, respectively. These images are uniformly spread across 10 classes in CIFAR-10 and 200 randomly selected classes in ImageNet. For classification models, we adopt models from the RobustBench (Croce et al., 2021). For standardly trained models, WideResNet-28-10 (Zagoruyko & Komodakis, 2016) with 94.78% for CIFAR-10 and ResNet-50 (He et al., 2016) with 76.52% for ImageNet are used. For adversarially trained models, we use the WideResNet-28-10 model with 89.48% trained by Gowal et al. (2020) for CIFAR-10 and ResNet-50 model with 64.02% trained by Salman et al. (2020a) for ImageNet.

**Attack and Defense Methods.** We consider 5 SOTA query-based attacks for evaluation: NES (Ilyas et al., 2018), SimBA (Guo et al., 2019), Square (Andriushchenko et al., 2020), Boundary (Brendel et al., 2018), and HopSkipJump (Chen et al., 2020). Comprehensive descriptions and configurations of each attack can be found in Appendix D. The perturbation budget of $\ell_\infty$ attacks is set to 8/255 for CIFAR-10 and 4/255 for ImageNet. For $\ell_2$ attacks, the perturbation budget is set to 1.0 for CIFAR-10 and 5.0 for ImageNet. For defense mechanism, adversarially trained models are used as a strong robust baseline. Moreover, we include SOTA deterministic purification model DISCO Ho & Vasconcelos (2022) and spatial smoothing (Xu et al., 2018) for direct comparison. Finally, widely used random noise defense (Qin et al., 2021) serve as a baseline for introducing randomness. The detailed settings of each defense method are described in Appendix E. We report the robust accuracy of each defense method against each attack with 200/2500 queries for CIFAR-10/ImageNet.

Table 2: Evaluation results of MATT and other defense methods on CIFAR-10 and ImageNet under 5 SOTA query-based attacks. The robust accuracy under 200/2500 queries is reported. The best defense mechanism under 2500 queries are highlighted in bold and marked with gray.

| Datasets | Methods | Acc. | NES($\ell_\infty$) | SimBA($\ell_2$) | Square($\ell_\infty$) | Boundary($\ell_2$) | HopSkipJump($\ell_\infty$) |
|---|---|---|---|---|---|---|---|
| CIFAR-10 (WideResNet-28) | None | 100.0 | 88.0/12.6 | 51.8/3.16 | 28.1/0.90 | 93.2/62.0 | 75.9/76.6 |
| | AT | 90.2 | 88.4/83.1 | 87.5/79.0 | 81.7/71.0 | 89.3/**88.8** | 90.0/88.6 |
| | Smoothing | 80.6 | 53.5/8.00 | 45.0/5.50 | 6.00/0.00 | 74.6/42.0 | 68.1/66.1 |
| | Input Rand. | 81.4 | 78.8/75.0 | 75.4/69.6 | 67.6/63.8 | 78.5/80.6 | 74.9/78.2 |
| | DISCO | **91.1** | 87.6/36.6 | 82.0/16.6 | 22.1/2.20 | 88.7/69.6 | 85.4/86.2 |
| | MATT (**Ours**) | 88.7 | 88.8/**86.3** | 85.6/78.5 | 81.5/71.5 | 87.4/87.4 | 87.0/**89.4** |
| | MATT-AT (**Ours**) | 89.2 | 88.3/85.6 | 87.9/**85.0** | 86.0/**83.0** | 89.0/88.7 | 88.9/88.7 |
| ImageNet (ResNet-50) | None | 100.0 | 95.3/80.0 | 85.6/66.4 | 49.2/6.80 | 92.4/84.8 | 89.3/**86.3** |
| | AT | 75.2 | 68.2/66.8 | 71.2/66.3 | 69.2/61.2 | 74.7/74.5 | 75.1/74.9 |
| | Smoothing | **89.1** | 93.5/77.7 | 80.2/36.8 | 37.7/3.00 | 86.9/78.9 | 84.2/84.2 |
| | Input Rand. | 84.6 | 83.6/**82.4** | 80.6/76.2 | 81.4/**78.5** | 85.4/84.8 | 84.7/85.6 |
| | DISCO | 88.5 | 86.2/79.6 | 79.8/33.6 | 45.1/6.70 | 86.2/82.8 | 87.6/84.4 |
| | MATT (**Ours**) | 87.2 | 85.6/82.2 | 82.5/**80.1** | 85.0/77.2 | 87.0/**85.3** | 86.5/**86.3** |
| | MATT-AT (**Ours**) | 75.5 | 73.2/71.5 | 70.6/69.9 | 72.4 /69.5 | 74.2/74.7 | 74.3/73.3 |

## 5.2 Overall Defense Performance

Our numerical results on the effectiveness of the defense mechanisms are shown in Table 2.

**Clean Accuracy.** One major concern about performing input transformations is that they may compromise the accuracy of non-adversarial inputs. We observe that MATT achieves comparable clean accuracy to the standardly trained model on both CIFAR-10 and ImageNet. Moreover, we observe that MATT can be combined with adversarially trained models to improve its clean accuracy for free. Detailed information on the influence of MATT on clean accuracy can be found in Appendix I.

**Failure of Deterministic Purification.** As suggested in our theoretical analysis, deterministic transformations, face the risk of introducing extra vulnerability and accelerating the attacks. As shown in Table 2, spatial smoothing always accelerating the attack and DISCO suffer from a significant drop in robust accuracy under 2500 queries against strong attack (Square Attack). These results reflect the importance of introducing randomness in purification.

**Effectiveness of MATT.** Our mechanism, built upon adversarial purification, achieves moderate clean accuracy to be the highest or the second to be the highest. This desired property allows for getting trustworthy results. Moreover, it achieves the highest robust accuracy under 2500 queries on CIFAR-10 and ImageNet almost for all the attacks. Surprisingly, the random noise defense performs well on natural images from ImageNet datasets, while MATT achieves a comparable results when random noise defense achieved the best results. This suggests that MATT can be used as a general defense mechanism for both $\ell_\infty$ and $\ell_2$ attacks.

## 6 Conclusion

This paper introduces a novel theory-backed image purification mechanism utilizing local implicit function to defend deep neural networks against query-based adversarial attacks. The mechanism enhances classifier robustness and reduces successful attacks whilst also addressing vulnerabilities of deterministic transformations. Its effectiveness and robustness, which increase with the addition of purifiers, have been authenticated via extensive tests on CIFAR-10 and ImageNet. Our work highlights the need for dynamic and efficient defense mechanisms in machine learning systems.

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

## A   BACKGROUND FOR GENERAL DEFENSE

**More on General Defense for Query-based Attacks** The defense for black-box query-based attacks remain relatively unexplored compared to the defense for white-box methods Qin et al. (2021). Under considerations of real-world constraints such as clean accuracy and inference speed, most

training-time and testing-time defenses present significant limitations for deployment in real-world MLaaS systems. For training-time defense, the aim is to improve the worst-case robustness of the models. Adversarial training (AT) has been considered as one of the fundamental practices of training time defense, where models are trained on augmented datasets with specially-crafted samples to ensure robustness of the models (Madry et al., 2018). Other training-time examples like gradient masking (Tramèr et al., 2018) and defensive distillation Papernot & McDaniel (2016) are also proposed to improve the robustness of the models. Nonetheless, such methods are unsuitable for MLaaS systems because of the extensive training costs and potential for decreased accuracy on clean examples. With regards to testing-time defense, a prominent defense from white-box attacks, randomized smoothing, can ensure the robustness of the model within a certain confidence level (Yang et al., 2020), a feature known as certified robustness. Another example for multiple inference to improve the robustness is called random self-ensemble Liu et al. (2018). However, the inference speed of randomized smoothing is too slow to be deployed on real-world MLaaS systems. Other testing-time defenses tend towards randomization of the input or output. Rand noise defense proposed by Qin et al. (2021) leverages Gaussian noises as the input to the model to disturb the gradient estimation. Yet, the defense is ineffective against strong attack methods and hurts the clean accuracy. The output-based defense, like confidence poisoning (Chen et al., 2022) influences the examples on the classification boundary and cannot defend against the decision-based attacks.

## B    SEARCH TECHNIQUES FOR BLACK-BOX ATTACKS

**Projected Gradient Descent** A common approach of performing adversarial attacks (often white-box) is to leverage projected gradient descent algorithm (Carlini & Wagner, 2017):

$$\boldsymbol{x}_{t+1} = Proj_{\mathcal{N}_R(\boldsymbol{x})}(\boldsymbol{x} - \eta_t g(\boldsymbol{x})_t). \tag{10}$$

**Gradient Estimation** While there can be various gradient estimators, we consider the following gradient estimator in our theoretical analysis:

$$g(\boldsymbol{x}) = \frac{f(\boldsymbol{x} + \mu\mathbf{u}) - f(\boldsymbol{x})}{\mu}\mathbf{u}. \tag{11}$$

**Heuristic Search** For heuristic search, the main issue is to determine the search direction. One commonly used search direction can be:

$$s(\boldsymbol{x}) = \mathbb{I}(h(\boldsymbol{x}) < 0) \cdot \mu\mathbf{u}, \quad \text{where} \quad h(\boldsymbol{x}) = f(\boldsymbol{x} + \mu\mathbf{u}) - f(\boldsymbol{x}), \tag{12}$$

where $\mathbb{I}$ is the indicator function. The search direction is determined by the sign of the objective function. If the objective function is negative, the search direction is the gradient direction. Otherwise, the search direction is the opposite of the gradient direction. The corresponding updating direction will be Equation (10) with $-\eta_t g(\boldsymbol{x})_t$ replaced by $s(\boldsymbol{x}_t)$.

## C    COMPARISON OF THE INFERENCE SPEED

We have tested the inference speed of DISCO and MATT using a workstation with a single NVIDIA RTX 4090 GPU. We set the batch size of inference to be 1 and vary the number of the purifiers from 1 to 10. We record the time of the last 900 inferences of a total of 1000 iterations. Moreover, to ensure a fair comparison on the inference mechanism, we use the same encoder and decoder for DISCO and MATT so that they only differ in the way of inference. Note that any software and services that may affect the test has been turned off. We use the same standardly and adversarially trained models as section 5 as the baseline of the comparison. The results for CIFAR-10 and ImageNet datasets are shown in Figure 7 and Figure 8, respectively. While our mechanism does not increase inference cost with the increase of the purifiers, the inference cost of DISCO almost increase linearly. Moreover, we have noticed that on ImageNet datasets, the inference speed of MATT is even faster than the baseline model, which makes our mechanism suitable for real-world MLaaS systems.

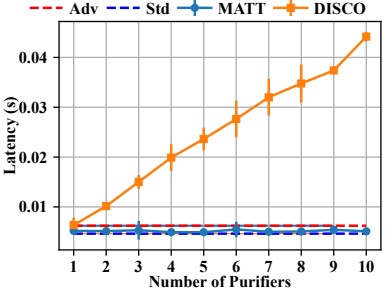

Figure 7: The inference speed of DISCO and MATT on CIFAR-10 dataset.

Figure 8: The inference speed of DISCO and MATT on ImageNet dataset.

## D  DETAILS OF THE ATTACKS

We utilize 5 SOTA query-based attacks for evaluation: NES (Ilyas et al., 2018), SimBA (Guo et al., 2019), Square (Andriushchenko et al., 2020), Boundary (Brendel et al., 2018), and Hop-SkipJump (Chen et al., 2020). The category of them is listed below in Table 3.

Table 3: The category of the attacks along with the techniques they use.

|  | Gradient Estimation | Random Search |
|---|---|---|
| Score-based | NES | Square, SimBA |
| Decision-based | Boundary | HopSkipJump |

**Implementation** For Boundary Attack and HopSkipJump Attack, we adopt the implementation from Foolbox (Rauber et al., 2020). For Square Attack and SimBA, we use the implementation from ART library (Nicolae et al., 2018). For NES, we implement it under the framework of Foolbox.

**Hyperparameters** The hyperparameters used for the attacks are listed below for full reproducibility.

Table 4: The hyperparameters used for NES.

|  | CIFAR-10 | ImageNet |
|---|---|---|
| $\eta$ (learning rate) | 0.01 | 0.0005 |
| $q$ (number of points used for estimation) | 100 | 100 |

Table 5: The hyperparameters used for SimBA.

|  | CIFAR-10 | ImageNet |
|---|---|---|
| $\eta$ (step size) | 0.2 | 0.2 |

Table 6: The hyperparameters used for Square.

|  | CIFAR-10 | ImageNet |
|---|---|---|
| $\mu$ (Fraction of Pixel Changed) | $0.05 \sim 0.5$ | $0.05 \sim 0.5$ |

Table 7: The hyperparameters used for Boundary Attack.

|  | CIFAR-10 | ImageNet |
|---|---|---|
| $\eta_{sph}$ (Spherical Step) | 0.01 | 0.01 |
| $\eta_{src}$ (Source Step) | 0.01 | 0.01 |
| $\eta_c$ (Source Step Converge) | 1E-7 | 1E-7 |
| $\eta_a$ (Step Adaptation) | 1.5 | 1.5 |

Table 8: The hyperparameters used for HopSkipJump Attack.

|  | CIFAR-10 | ImageNet |
|---|---|---|
| $n$ (number for estimation) | 100 | 100 |
| $\gamma$ (Step Control Factor) | 1 | 1 |

## E  DETAILED INFORMATION FOR THE DEFENSE

We compare our algorithm with three types of baseline defense. For random noise defense, we use a Gaussian noise with $\sigma = 0.041$ as the input to the classifier. For the spatial smoothing transformations, we set the size of the kernel filter to be 3. For DISCO model, we implement a naive version without randomness using pre-trained models from the official implementation. We pick

the pre-trained model under PGD attack (Madry et al., 2018) as the core local implicit model for DISCO.

## F    DETAILS FOR MATT

### F.1    EFFICIENT STRUCTURE FOR INFERENCE SPEED UP

After we delve into the local implicit function implementation Chen et al. (2021), we find that the local ensemble mechanism that used to improve the performance of the local implicit function in super-resolution tasks is not necessary for our task. A conceptual illustration of this technique is shown in Figure 9. As image purification is a one-to-one pixel mapping task, the local ensemble mechanism inference the same pixel four times and take the average value, which is meaningless for our task. Therefore, we remove this mechanism and only use the local implicit function to purify the image. This modification accelerates the inference speed by a factor of four.

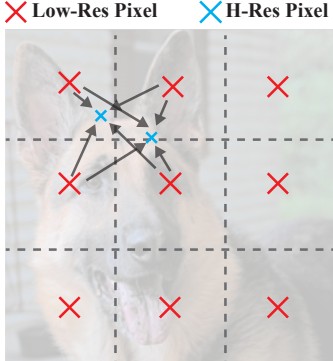

Figure 9: An illustration of the local ensemble mechanism in the local implicit function for multi-resolution support. High resolution pixels are predicted based on high-level features from nearby low resolution pixels.

### F.2    TRAINING DIVERSIFIED PURIFIERS

For improving the diversity of the purifiers, we consider the following influence factors in Table 9 and use their combinations to train 12 different purification models for each dataset. For CIFAR-10, we use a pre-trained ResNet-18 model (Dadalto, 2022) for generating adversarial examples. For ImageNet, we use a pre-trained ResNet-50 model (Torchvision, 2023) for generating adversarial examples.

Table 9: The factors that are considered when training diversified models for MATT.

| Hyperparameter | Value |
|---|---|
| Attack Tyep | FGSM (Goodfellow et al., 2015) |
| | PGD (Madry et al., 2018) |
| | BIM Kurakin et al. (2017) |
| Encoder Structure | RCAN (Zhang et al., 2018), EDSR |
| Feature Depth | 32, 64 |

We diversify the purifiers from the following aspects:

- **Structural Diversity:** By leveraging different structures of encoders and decoders, we can diversify the purifiers.
- **Random Patch-wise Diversity:** By performing random patch-wise purification, we are actually combining different purifiers to form a new purifier.

# G SUPPLEMENTARY MATERIALS FOR THEORETICAL ANALYSIS

## G.1 IMPORTANT DEFINITIONS

**Definition 1.** *The Gaussian-Smoothing function corresponding to $f(\boldsymbol{x})$ with $\mu > 0, \mathbf{u} \sim \mathcal{N}(\mathbf{0}, \boldsymbol{I})$ is*

$$f_\mu(\boldsymbol{x}) = \frac{1}{(2\pi)^{d/2}} \int f(\boldsymbol{x} + \mu\mathbf{u}) e^{-\frac{\|\mathbf{u}\|^2}{2}} \, \mathrm{d}\mathbf{u} \tag{13}$$

## G.2 PROOF OF THEOREM 1

The essential lemmas are given as follows, the complete proofs are shown in Nesterov & Spokoiny (2017).

**Lemma 1.** *Let $f(\boldsymbol{x})$ be the Lipschitz-continuous function, $|f(\boldsymbol{y}) - f(\boldsymbol{x})| \leq L_0(f)\|\boldsymbol{y} - \boldsymbol{x}\|$. Then*

$$L_1(f_\mu) = \frac{d^{\frac{1}{2}}}{\mu} L_0(f)$$

We define the $p$-order moment of normal distribution as $M_p$. Then we have

**Lemma 2.** *For $p \in [0, 2]$, we have*

$$M_p \leq d^{\frac{p}{2}}$$

*If $p \geq 2$, the we have two-side bounds*

$$d^{\frac{p}{2}} \leq M_p \leq (p+d)^{\frac{p}{2}}$$

**Lemma 3.** *Let $f(\boldsymbol{x})$ be the Lipschitz-continuous function, $|f(\boldsymbol{y}) - f(\boldsymbol{x})| \leq L_0(f)\|\boldsymbol{y} - \boldsymbol{x}\|$. And $\boldsymbol{m}(\boldsymbol{x})$ is Lipschitz-continuous for every dimension. Then*

$$L_0(f \circ \boldsymbol{m}) \leq L_0(f)L_0(\boldsymbol{m})$$

*where $L_0(\boldsymbol{m})$ is defined as $L_0(\boldsymbol{m}) = \max_i L_0(m_i)$.*

*Proof.*

$$|f(\boldsymbol{m}(\boldsymbol{y})) - f(\boldsymbol{m}(\boldsymbol{x}))| \leq L_0(f)\|\boldsymbol{m}(\boldsymbol{y}) - \boldsymbol{m}(\boldsymbol{x})\|$$

$$= L_0(f)\sqrt{\sum_{i=1}^{d} L_0(\boldsymbol{m}_i)^2 (\boldsymbol{y}_i - \boldsymbol{x}_i)^2} \tag{14}$$

$$\leq L_0(f)L_0(\boldsymbol{m})\|\boldsymbol{y} - \boldsymbol{x}\|$$

$\square$

This is the proof for Theorem 1.

*Proof.* According to the property of Lipschitz-continuous gradient,

$$F_{\mu,K}(\boldsymbol{x}_{t+1}) \leq F_{\mu,K}(\boldsymbol{x}_t) - \eta_t \langle \nabla F_{\mu,K}(\boldsymbol{x}_t), G_{\mu,K}(x_t) \rangle + \frac{1}{2}\eta_t^2 L_1(F_{\mu,K})\|G_{\mu,K}(\boldsymbol{x}_t)\|^2 \tag{15}$$

The $G_{\mu,K}(\boldsymbol{x}_t)$ can be decomposed as

$$G_{\mu,K}(\boldsymbol{x}_t) = \frac{f(\boldsymbol{m}_{k_{t1}}(\boldsymbol{x} + \mu\mathbf{u})) - f(\boldsymbol{m}_{k_{t2}}(\boldsymbol{x}))}{\mu}\mathbf{u}_t$$

$$= \frac{f(\boldsymbol{m}_{k_{t1}}(\boldsymbol{x} + \mu\mathbf{u})) - f(\boldsymbol{m}_0(\boldsymbol{x} + \mu\mathbf{u})) + f(\boldsymbol{m}_0(\boldsymbol{x} + \mu\mathbf{u})) - f(\boldsymbol{m}_0(\boldsymbol{x}))}{\mu}\mathbf{u}_t \tag{16}$$

$$+ \frac{f(\boldsymbol{m}_0(\boldsymbol{x})) - f(\boldsymbol{m}_{k_{t2}}(\boldsymbol{x}))}{\mu}\mathbf{u}_t$$

The squared term $\|G_{\mu,K}(\boldsymbol{x}_t)\|^2$ is bounded by

$$\|G_{\mu,K}(\boldsymbol{x}_t)\|^2 \leq \frac{4\nu^2}{\mu^2}L_0(f)^2\|\mathbf{u}_t\|^2 + \frac{4\nu}{\mu}L_0(F)L_0(f)\|\mathbf{u}_t\|^3 + L_0(F)^2\|\mathbf{u}_t\|^4 \tag{17}$$

Take the expectation over $\mathbf{u}_t$, $k_{t1}$, and $k_{t2}$, use Lemma 2, we have

$$
\begin{aligned}
F_{\mu,K}(\boldsymbol{x}_{t+1}) \leq\ & F_{\mu,K}(\boldsymbol{x}_t) - \eta_t \|\nabla F_{\mu,K}(\boldsymbol{x}_t)\|^2 \\
& + \frac{1}{2}\eta_t^2 L_1(F_{\mu,K})(\frac{4\nu^2}{\mu^2}L_0(f)^2 d + \frac{4\nu}{\mu}L_0(F)L_0(f)(d+3)^{\frac{3}{2}} + L_0(F)^2(d+4)^2)
\end{aligned}
\tag{18}
$$

For $L_1(F_{\mu,K})$, we have:

$$
L_1(F_{\mu,K}) = \frac{1}{K}\sum_{k=1}^{K} L_1(f_\mu(\boldsymbol{m}_k)) \leq \frac{L_0(F)d^{\frac{1}{2}}}{\mu}
\tag{19}
$$

Use Lemma 1, and the dimension $d$ is high, we have

$$
\begin{aligned}
F_{\mu,K}(\boldsymbol{x}_{t+1}) \leq\ & F_{\mu,K}(\boldsymbol{x}_t) - \eta_t \|\nabla F_{\mu,K}(\boldsymbol{x}_t)\|^2 \\
& + \frac{1}{2}\eta_t^2 \frac{L_0(f)^3 L_0(\boldsymbol{m}_0)d^{\frac{3}{2}}}{\mu}(\frac{4\nu^2}{\mu^2} + \frac{4\nu}{\mu}L_0(\boldsymbol{m}_0)d^{\frac{1}{2}} + L_0(\boldsymbol{m}_0)^2 d)
\end{aligned}
\tag{20}
$$

We take the expectation over $\mathbf{U}_t, \mathrm{k}_t$.

$$
\begin{aligned}
\mathbb{E}_{\mathbf{U}_t,\mathrm{k}_t}[F_{\mu,K}(\boldsymbol{x}_{t+1})] \leq\ & \mathbb{E}_{\mathbf{U}_{t-1},\mathrm{k}_{t-1}}[F_{\mu,K}(\boldsymbol{x}_t)] - \eta_t \mathbb{E}_{\mathbf{U}_t,\mathrm{k}_t}[\|\nabla F_{\mu,K}(\boldsymbol{x}_t)\|^2] \\
& + \frac{1}{2}\eta_t^2 \frac{L_0(f)^3 L_0(\boldsymbol{m}_0)d^{\frac{3}{2}}}{\mu}(\frac{4\nu^2}{\mu^2} + \frac{4\nu}{\mu}L_0(\boldsymbol{m}_0)d^{\frac{1}{2}} + L_0(\boldsymbol{m}_0)^2 d)
\end{aligned}
\tag{21}
$$

Now consider constant step size $\eta_t = \eta$, and sum over $t$ from 0 to $Q$, we have

$$
\begin{aligned}
\frac{1}{Q+1}\sum_{t=0}^{Q} \mathbb{E}_{\mathbf{U}_t}[\|\nabla F_{\mu,K}(\boldsymbol{x}_t)\|^2] \leq\ & \frac{1}{\eta}(\frac{F_{\mu,K}(\boldsymbol{x}_0) - F_K^*}{Q+1}) \\
& + \frac{1}{2}\eta \frac{L_0(f)^3 L_0(\boldsymbol{m}_0)d^{\frac{3}{2}}}{\mu}(\frac{4\nu^2}{\mu^2} + \frac{4\nu}{\mu}L_0(\boldsymbol{m}_0)d^{\frac{1}{2}} + L_0(\boldsymbol{m}_0)^2 d)
\end{aligned}
\tag{22}
$$

Since the distance between the input variable should be bounded by $R$ and use Lipschitz-continuous, we have

$$
\|F_{\mu,K}(\boldsymbol{x}_0) - F_K^*\| \leq \frac{1}{K}L_0(f)\sum_{k=0}^{K} L_0(\boldsymbol{m}_k)R \leq L_0(F)R
\tag{23}
$$

Considering bounded $\mu \leq \hat{\mu} = \frac{\epsilon}{d^{\frac{1}{2}}L_0(F)}$ to ensure local Lipschitz-continuity, and set $\gamma(\boldsymbol{m}_0,\nu) = \frac{4\nu^2}{\mu^2} + \frac{4\nu}{\mu}L_0(\boldsymbol{m}_0)d^{\frac{1}{2}} + L_0(\boldsymbol{m}_0)^2 d$

$$
\frac{1}{Q+1}\sum_{t=0}^{Q} \mathbb{E}_{\mathbf{U}_t,\mathrm{k}_t}[\|\nabla F_\mu(\boldsymbol{x}_t)\|^2] \leq \frac{1}{\eta}(\frac{L_0(F)R}{Q+1}) + \frac{1}{2}\eta\frac{L_0(f)^4 L_0(\boldsymbol{m}_0)^2}{\epsilon}d^2\gamma(\boldsymbol{m}_0,\nu)
\tag{24}
$$

Minimize the right hand size,

$$
\eta = \sqrt{\frac{2R\epsilon}{(Q+1)L_0(f)^3 d^2}} \cdot \sqrt{\frac{1}{L_0(\boldsymbol{m}_0)\gamma(\boldsymbol{m}_0,\nu)}}
\tag{25}
$$

And we get

$$
\frac{1}{Q+1}\sum_{t=0}^{Q} \mathbb{E}_{\mathbf{U}_t,\mathrm{k}_t}[\|\nabla F_\mu(\boldsymbol{x}_t)\|^2] \leq \sqrt{\frac{2L_0(f)^5 R d^2}{(Q+1)\epsilon}} \cdot \sqrt{\gamma(\boldsymbol{m}_0,\nu)L_0(\boldsymbol{m}_0)^3}
\tag{26}
$$

To guarantee the expected squared norm of the gradient of function $F_\mu$ of the order $\delta$, the lower bound for the expected number of queries is

$$
O(\frac{L_0(f)^5 R d^2}{\epsilon\delta^2}\gamma(\boldsymbol{m}_0,\nu)L_0(\boldsymbol{m}_0)^3)
\tag{27}
$$

$\square$

*Proof.*

$$P(Sign(H(\boldsymbol{x})) \neq Sign(H_K(\boldsymbol{x}))) \leq P(|H_K(\boldsymbol{x}) - H(\boldsymbol{x})| \geq |H(x)|)$$

$$\leq \frac{\mathbb{E}[|H_K(\boldsymbol{x}) - H(\boldsymbol{x})|]}{|H(\boldsymbol{x})|}$$

$$\leq \frac{\sqrt{\mathbb{E}[(H_K(\boldsymbol{x}) - H(\boldsymbol{x})])^2}}{|H(\boldsymbol{x})|}$$

$$\leq \frac{\sqrt{\mathbb{E}[2(f(\boldsymbol{m}_{k_1}(\boldsymbol{x} + \mu\mathrm{u})) - f(\boldsymbol{m}_0(\boldsymbol{x} + \mu\mathrm{u})))^2 + 2(f(\boldsymbol{m}_{k_2}(\boldsymbol{x})) - f(\boldsymbol{m}_0(\boldsymbol{x})))^2]}}{|H(\boldsymbol{x})|}$$

$$\leq \frac{2\nu L_0(f)}{|H(\boldsymbol{x})|}$$

$$(28)$$

□

# H    ACCURACY CONVERTED USING THE BASE ACCURACY

As we have evaluated the performance of MATT on the *correctly* classified examples, the resulting robust accuracy on these examples may seem higher than the test accuracy reported in other works. To avoid confusion and ease the comparison of our work with other works, we have provided a table of converted accuracy (original robust accuracy multiplies the base accuracy of the model) in Table 10.

Table 10: The converted accuracy using results from Table 2.

| Datasets | Methods | Acc. | NES($\ell_\infty$) | SimBA($\ell_2$) | Square($\ell_\infty$) | Boundary($\ell_2$) | HopSkipJump($\ell_\infty$) |
|---|---|---|---|---|---|---|---|
| | None | 94.8 | 83.4/11.9 | 49.1/3.0 | 26.6/0.9 | 88.4/60.0 | 72.0/72.6 |
| | AT | 85.5 | 83.8/78.8 | 83.0/74.9 | 77.5/67.3 | **84.7/84.2** | 85.3/84.0 |
| CIFAR-10 | Smoothing | 76.4 | 50.7/7.6 | 42.7/5.2 | 5.7/0.0 | 70.7/39.8 | 64.6/62.7 |
| (WideResNet-28) | Input Rand. | 77.1 | 74.7/71.1 | 71.5/66.0 | 64.1/60.5 | 74.4/76.4 | 71.0/74.1 |
| | DISCO | **86.3** | 83.0/34.7 | 77.7/15.7 | 21.0/2.1 | 84.1/66.0 | 81.0/81.7 |
| | MATT (**Ours**) | 84.1 | **84.2/81.8** | 81.1/74.4 | 77.3/67.8 | 82.9/82.9 | 82.5/84.8 |
| | MATT-AT (**Ours**) | 84.6 | 83.7/81.1 | **83.3/80.6** | **81.5/78.7** | 84.4/84.1 | 84.3/84.1 |
| | None | 76.5 | 72.9/61.2 | 65.5/50.8 | 37.6/5.2 | 70.7/64.9 | **68.3/66.0** |
| | AT | 57.5 | 52.2/51.1 | 54.5/50.7 | 52.9/46.8 | 57.1/57.0 | 57.5/57.3 |
| ImageNet | Smoothing | **68.2** | 71.5/59.4 | 61.4/28.2 | 28.8/2.3 | 66.5/60.4 | 64.4/64.4 |
| (ResNet-50) | Input Rand. | 64.7 | **64.0/63.0** | 61.7/58.3 | 62.3/60.1 | 65.3/64.9 | 64.8/65.5 |
| | DISCO | 67.7 | 65.9/60.9 | 61.0/25.7 | 34.5/5.1 | 65.9/63.3 | 67.0/64.6 |
| | MATT (**Ours**) | 66.7 | 65.5/62.9 | **63.1/61.3** | 65.0/59.1 | **66.6/65.3** | 66.2/66.0 |
| | MATT-AT (**Ours**) | 57.8 | 56.0/54.7 | 54.0/53.5 | 55.4/53.2 | 56.8/57.1 | 56.8/56.1 |

# I    INFLUENCE ON CLEAN ACCURACY

One of the biggest advantage of local implicit purification is that it does not affect the clean accuracy of the model. While the results for evaluation of our mechanism's robust accuracy are shown in Table 2 in section 5, we also provide the results for clean accuracy in figure 10. Moreover, we have conducted extra experiments on the influence of the numbers of the image patches on the clean accuracy. The results are shown in figure 11. The results are obtained using the whole test set of CIFAR-10 and validation set of ImageNet.

**Comaprison of Defense Mechanisms.** We first test clean accuracy on each purification model contained in DISCO and our method. The label name refers to the white-box attacks used to generate adversarial examples for training the purification model. For MATT, a list of the purification model and their according attack and encoder combination can be found in Table 11. For both datasets, all the purification models have a better clean accuracy than adding random noise. Moreover, they all achieve better clean accuracy than adversarially trained models on ImageNet dataset.

**Influence of the Number of Patches.** We then test the influence of the number of patches on the clean accuracy. In MATT, we only use image patches for feature encoding and purification.

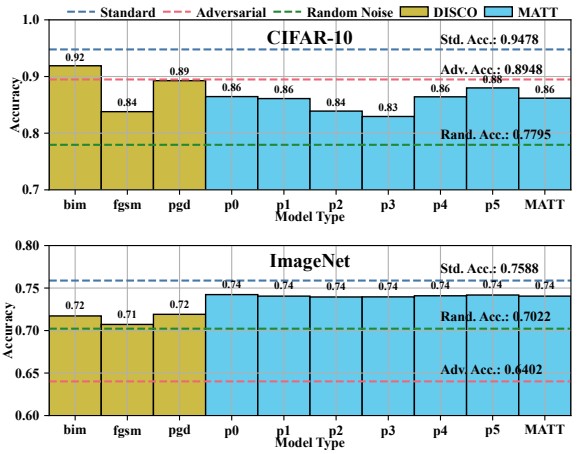
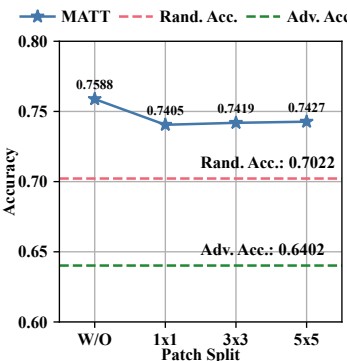

Figure 10: Comparison of defense mechanisms and models on clean accuracy. **Upper Figure:** CIFAR-10 dataset. **Lower Figure:** ImageNet dataset.

Figure 11: Influence of the number of image patches in MATT.

Therefore, the number of patches is a hyperparameter that can be tuned. We test the influence of the number of patches on the clean accuracy. The results are shown in figure 11. We can see that the clean accuracy is not affected by the number of patches.

Table 11: The purification model used in MATT.

| Model Type | p0 | p1 | p2 | p3 | p4 | p5 |
|---|---|---|---|---|---|---|
| Attack_Encoder | BIM_EDSR | BIM_RCAN | FGSM_EDSR | FGSM_RCAN | PGD_EDSR | PGD_RCAN |

