# OpenReview forum: "MATT: Random Local Implicit Purification for Defending Query-based Attacks"
_ICLR.cc/2024/Conference — Submitted to ICLR 2024_

### Official Review · Reviewer_TG1v · 2023-10-30

**Soundness:** 2 fair
**Presentation:** 2 fair
**Contribution:** 2 fair
**Rating:** 3
**Confidence:** 4

**Summary:**

It proposes MATT, that employs random patch-wise purifications with an ensemble of lightweight purification models. These models leverage
the local implicit function and rebuild the natural image manifold with low inference latency.

**Strengths:**

It provides a theoretical analysis on the effectiveness of the proposed purifier-based defense mechanism based on the convergence of black-box attacks. The theoretical analysis points out the potential vulnerabilities of deterministic transformation functions and suggests
the robustness of the system increase with the number of purifiers.

**Weaknesses:**

The presentation can be improved. There are some typos, such as 'CICAR-10 and ImageNet' above the conclusion section. It introduces the  PRELIMINARIES in section 3 with some equations. But it seems that section 3 is not necessary as the equations are not used and the attacks are easy to follow. Since it puts many information to the appendix, maybe it is better to move some important information such as experimental results to the main paper instead of section 3.

There may be some problems with the experimental setting. In table 2, without defense (the 'None' row),  some attacks does not achieve high successful attack rate with low robust accuracy. For example, the robust accuracy of NES for unprotected model is 80%, similarly, Boundary with 84.8%, HopSkipJump with 86.3% on Imagenet. It means that without any defenses, the attacks are quite weak. The setting may have some problems so that the attacks does not perform well. Comparing with these weak attacks due to inappropriate setting, its better performance does not mean that the proposed method is really better. And sometimes the proposed method is not the best compared the baselines. The experiments do not seem to be solid.

For black-box models, transfer attack is often adopted an attack method, to generate adversarial examples with white-box substitute model and transfer the examples to attack the black-box model. It is better to test the performance under transfer attack.

It mainly discusses the black-box model. But the method is general and it can be applied for white-box models. What is its performance if the model is white-box? It is better to provide more insights for the proposed method.

It mentions that the proposed method is more efficient and can achieve some speedup. Maybe it is better to provide some detailed complexity computation or discussion to help better understanding why it is more efficient. Besides, it seems that it needs to train multiple purifiers. It may add some additional training efforts and make the training more complex. It is better to discuss the training complexity.

It is better to experiment with more architectures. Currently it only shows the results of WideResNet-28-10 for CIFAR and ResNet-50 for ImageNet. More results on other architectures can help better demonstrate the generalization of the proposed method.

**Questions:**

see the weakness.

---

> ### Author Response · Authors · 2023-11-19
>
> We thank reviewer TG1v for the thoughtful and detailed review. We have made necessary amendments to our paper according to the feedback. Please find below our responses to your comments.
>
> ## Presentation of the PRELIMINARIES
> We provide the necessary equations in the *Preliminaries* section, especially the new gradient approximator $g(x)$, because we believe they can facilitate readers to better understand our subsequent theoretical analysis.
>
> We agree and thank the reviewer's suggestions for the balance of detail and brevity, and we thus decide to move the detailed equations to the *Appendix* while retaining the information regarding the adversarial purification in the main body due to its novelty in the black-box attacks research community.
>
> ## Problem with Experimental Setting
> We regret that the reviewer may misunderstand our experimental settings, and we wish to clarify and justify our settings to address any potential misunderstandings.
>
> * We did not see our evaluation procedures as problematic because we follow the settings of a related work [1] that uses **correctly** classified examples instead of all test samples to **exclude** the influence of misclassification on the attack or the defense performance.
>
> * This experimental setting leads to the seemingly high accuracy of **80%, 84.8%, and 86.3%** on the ImageNet dataset. In fact, the corresponding accuracy on the whole dataset is **60%, 63.6%, and 64.7%**, respectively, assuming the original accuracy is **75%** and no statistical bias with randomly chosen examples.
>
> * We have provided a table of **converted** accuracy below and in *Appendix H* of the revised paper for your reference.
>
> Table 1: A table of converted accuracy.
> | Datasets | Methods       | Acc. | NES       | SimBA     | Square    | Boundary  | HopSkipJump |
> | -------- | ------------- | ---- | --------- | --------- | --------- | --------- | ----------- |
> | CIFAR-10 | None          | 94.8 | 83.4/11.9 | 49.1/3.0  | 26.6/0.9  | 88.4/60.0 | 72.0/72.6   |
> |          | AT            | 85.5 | 83.8/78.8 | 83.0/74.9 | 77.5/67.3 | 84.7/84.2 | 85.3/84.0   |
> |          | Smoothing     | 76.4 | 50.7/7.6  | 42.7/5.2  | 5.7/0.0   | 70.7/39.8 | 64.6/62.7   |
> |          | InputRand.    | 77.1 | 74.7/71.1 | 71.5/66.0 | 64.1/60.5 | 74.4/76.4 | 71.0/74.1   |
> |          | DISCO         | 86.3 | 83.0/34.7 | 77.7/15.7 | 21.0/2.1  | 84.1/66.0 | 81.0/81.7   |
> |          | MATT(Ours)    | 84.1 | 84.2/81.8 | 81.1/74.4 | 77.3/67.8 | 82.9/82.9 | 82.5/84.8   |
> |          | MATT-AT(Ours) | 84.6 | 83.7/81.1 | 83.3/80.6 | 81.5/78.7 | 84.4/84.1 | 84.3/84.1   |
> | ImageNet | None          | 76.5 | 72.9/61.2 | 65.5/50.8 | 37.6/5.2  | 70.7/64.9 | 68.3/66.0   |
> |          | AT            | 57.5 | 52.2/51.1 | 54.5/50.7 | 52.9/46.8 | 57.1/57.0 | 57.5/57.3   |
> |          | Smoothing     | 68.2 | 71.5/59.4 | 61.4/28.2 | 28.8/2.3  | 66.5/60.4 | 64.4/64.4   |
> |          | InputRand.    | 64.7 | 64.0/63.0 | 61.7/58.3 | 62.3/60.1 | 65.3/64.9 | 64.8/65.5   |
> |          | DISCO         | 67.7 | 65.9/60.9 | 61.0/25.7 | 34.5/5.1  | 65.9/63.3 | 67.0/64.6   |
> |          | MATT(Ours)    | 66.7 | 65.5/62.9 | 63.1/61.3 | 65.0/59.1 | 66.6/65.3 | 66.2/66.0   |
> |          | MATT-AT(Ours) | 57.8 | 56.0/54.7 | 54.0/53.5 | 55.4/53.2 | 56.8/57.1 | 56.8/56.1   |
>
> [1] Vo V, Abbasnejad E M, Ranasinghe D. QUERY EFFICIENT DECISION BASED SPARSE ATTACKS AGAINST BLACK-BOX DEEP LEARNING MODELS International Conference on Learning Representations. 2021.

---

> ### Author Response · Authors · 2023-11-19
>
> ## Performance on Transfer Attack and White-box Attack
> * We did not provide the performance of transfer attacks and white-box attacks because they are out of the scope of this work that aims to provide a general defense mechanism for **query-based attacks**.
>
> * But our method can surely be used to defend against **white-box attacks** and **transfer attacks** according to our theoretical analysis on **gradient estimation**.
>
> ## Efficiency and Training Complexity
> * We have achieved a **constant** inference cost compared to **linearly** increase cost  for ensembling. Moreover, we have achieved a **'4x'** inference speed up by removing the local ensemble technique. As for training complexity, it has a **constant** factor difference。
>
> * We have already discussed the efficiency of from **linearly increase** to **constant** in *Section 4.2*. This efficiency is further validated by experiments in *Appendix C*.
>
> * Another **'4x'** inference speedup is discussed in *Section 4.2* and *Appendix F.1*.
>
> * Training complexity has a **constant** factor difference with training one purifier due to the presence of multiple purification models.
>
> ## Generalization Ability
> * The generalization between different model structures has already been included in the **training** and **testing** process of our purifiers. Specifically, we train the purifiers using adversarial examples **generated** by Table 2, while we test the purifiers on models **listed** in Table 3. This training and testing gap shows that our purifier can generalize to protect **unseen** models.
>
> * **Training:** Generating adversarial examples with models listed in Table 2.
>
> * **Testing:** Evaluating the performance of our mechanism on models listed in Table 3.
>
> Table 2: Details of the models used for training purifiers.
> |     Model     | Accuracy | Dataset  |
> | :-----------: | :------: | :------: |
> | ResNet-18 [2] |  94.0%   | CIFAR-10 |
> | ResNet-50 [3] |  80.9%   | ImageNet |
>
> Table 3: Details of the models used for testing. The first two models are standardly trained model, the last two are adversarially trained model.
> |       Model       | Accuracy | Dataset  |
> | :---------------: | :------: | :------: |
> | WideResNet-28 [4] |  94.78%  | CIFAR-10 |
> |   ResNet-50 [5]   |  76.52%  | ImageNet |
> | WideResNet-28 [6] |  89.48%  | CIFAR-10 |
> |   ResNet-50 [7]   |  64.02%  | ImageNet |
>
> [2] Eduardo Dadalto. Resnet18 trained on cifar10. https://huggingface.co/edadaltocg/resnet18_cifar10, 2022. Accessed: 2023-07-01.
>
> [3] Torchvision. Resnet50 - torchvision main documentation. https://pytorch.org/vision/main/models/generated/torchvision.models.resnet50.html, 2023. Ac-
> cessed: 2023-11-11
>
> [4] Sergey Zagoruyko and Nikos Komodakis. Wide residual networks. In Proceedings of the British Machine Vision Conference 2016, (BMVC), 2016.
>
> [5] Francesco Croce, Maksym Andriushchenko, Vikash Sehwag, Edoardo Debenedetti, Nicolas Flammarion, Mung Chiang, Prateek Mittal, and Matthias Hein. Robustbench: a standardized adversarial robustness benchmark. In Proceedings of the Neural Information Processing Systems Track on Datasets and Benchmarks 1, (NeurIPS), 2021
>
> [6] Sven Gowal, Chongli Qin, Jonathan Uesato, Timothy A. Mann, and Pushmeet Kohli. Uncovering the limits of adversarial training against norm-bounded adversarial examples. CoRR, 2020
>
> [7] Hadi Salman, Andrew Ilyas, Logan Engstrom, Ashish Kapoor, and Aleksander Madry. Do adversarially robust imagenet models transfer better? In Advances in Neural Information Processing Systems 33: Annual Conference on Neural Information Processing Systems 2020, (NeurIPS), 2020a

---

> ### Author Response · Authors · 2023-11-21
>
> Thank you again for your time and effort in reviewing our work.
>
> There are only less than 2 days left to the rebuttal deadline, and we sincerely want to know whether our responses can successfully address all your concerns.
>
> Please also let us know if you have further concerns. We are glad to continually improve our work to address them.

---

### Official Review · Reviewer_cfuj · 2023-10-31

**Soundness:** 3 good
**Presentation:** 3 good
**Contribution:** 3 good
**Rating:** 6
**Confidence:** 3

**Summary:**

This paper introduces MATT, an efficient defense method that uses random patch-wise purifications with lightweight purification models, able to slow down query-based attacks' convergence and enhance classifier robustness. The approach is better than traditional defenses like adversarial training and gradient masking which are either computationally expensive or reduce the accuracy of non-adversarial inputs. Through theoretical verification and empirical experiments on CIFAR-10 and ImageNet, the paper confirm its effectiveness.

**Strengths:**

+ The paper is well written and easy to follow.
+ The theoretical contribution is effectively supports the claims made in the paper.
+ The empirical results are promising and indicate the potential effectiveness of the proposed defense mechanism.

**Weaknesses:**

- Some figures in the paper are unclear and difficult to understand, which can hinder readers' comprehension of the research.
- The defense mechanism and its components appear to be incremental improvements on existing methods rather than introducing a truly novel approach. It could benefit from a more innovative perspective.

**Questions:**

- Figure 3 lacks a clear explanation for the small batch of examples shown in (II). It would greatly benefit the readers if this could be clarified in the Figure.
- In Figure 4, there is a reference to "Attack during Training" phase, but it's not clear whether adversarial training is conducted. Could you please provide more details on this matter?
Generally, the figures used in the paper do not effectively convey the ideas or methods, and they need improvement to enhance the paper's accessibility.

- In Section 4.2, most of the design elements mentioned do not appear to offer significant contributions. They seem more like incremental improvements built upon the DISCO paper. For instance, the removal of positional encoding and local ensemble inference does not constitute a major innovative contribution, and some modifications in the "Random Patch-wise Purification" section seem more like workarounds than novel contributions.

Some typos:
  - there is no $\gamma$ in Eq (5) but the following description mentioned, I think it should be $\lambda$
  - CICAR-10 ==> CIFAR-10

---

> ### Author Response · Authors · 2023-11-19
>
> We deeply appreciate the comprehensive review and constructive feedback provided by reviewer cfuj on our work.  We hope the following explanations and modifications will address your concerns and improve the clarity of our paper.
>
> ## Revising the figures.
> We would like to express our gratitude to the reviewer for pointing out the issues in our figures. We have revised Figures 3 and 4 according to the suggestions. Additionally, we have also updated Figure 2 for a better illustration of our methods. Please find below our modifications to the figures.
>
> ### Figure 3
> * **Small batches:** The small batches are extracted from the original image, which is segmented into several patches. We have now added additional annotations to the figure to clarify this (*Cut up* operation).
> * **Ensembling:** The ensembling method encodes the entire image into high-level features using different purifiers. The final encoding is formed by randomly sampling from these features. We have improved the figure to better illustrate this process.
> * **Random Patch-wise Encoding:**  The original image is pre-segmented, and the patches are randomly assigned to different purifiers. The final encoding is formed by the high-level features of these patches.
>
> ### Figure 4
> * **Attack during Training:** This step aims to provide adversarial images for our training process. For better illustration, we have listed out the attack algorithms used (PGD, BIM, and FGSM).
> * **Revised Training Loop:** We have updated the training loop into a triangle shape to better illustrate the training process. The process involves generating adversarial images, using all purifiers for purification, and training the classifier with all the purified images under different combinations under $\ell_1$ loss. We have added these details to the figure.
> * **Omitting Testing Phase:** As our defense mechanism is an end-to-end process, we omitted the testing phase in the figure for brevity.
>
> ## Novelty and Contribution
> We acknowledge your concerns regarding the novelty of our work. We would like to clarify that our work's novelty lies in two main aspects:
> 1. **Theoretical Advancement:** Through rigorous theoretical analysis, we demonstrate that **a deterministic purification process** is unsuitable for black-box attack defense. This conclusion is also corroborated by our experimental findings (Robust accuracy **2.2%** for CIFAR-10, **6.70%** for ImageNet). This represents a significant theoretical advancement.
>
> 2. **Technical Contribution:** Stemming from this analysis, we introduce a random patch-wise mechanism with **constant inference cost** to enhance robustness, which signifies our technical contribution. Actually, we have addressed other efficiency issues (cancelling local ensemble, utilizing smaller encoders) for the existing purification methods but not listed as the main contribution in the paper. An acceleration from **linear time** to **constant time** is achieved in Figure 7 and Figure  8 compared to simple ensemble method, which enables the application of purification methods in real-world MLaaS systems (**million-fold extra cost**).
>
> We hope the observations from theoretical analysis will be instrumental in designing purification-based methods for black-box systems. We have emphasized these insights in our contribution section (Section 1) for further clarity in the revised version of the paper.

---

> ### Author Response · Authors · 2023-11-21
>
> Thank you again for your time and effort in reviewing our work.
>
> There are only less than 2 days left to the rebuttal deadline, and we sincerely want to know whether our responses can successfully address all your concerns.
>
> Please also let us know if you have further concerns. We are glad to continually improve our work to address them.

---

### Official Review · Reviewer_bHtq · 2023-11-01

**Soundness:** 3 good
**Presentation:** 3 good
**Contribution:** 3 good
**Rating:** 5
**Confidence:** 4

**Summary:**

The paper proposes a defense mechanism called MATT to address the threat of query-based black-box attacks. MATT employs random patch-wise purification and an ensemble of purification models. It slows down the convergence of query-based attacks and enhances the robustness of classifiers by combining randomness and purification. Extensive experiments confirm the effectiveness of the proposed method.

**Strengths:**

- The paper is skillfully composed, and I grasped its content effortlessly.
- The theoretical analysis is solid. The analysis of the Single Deterministic Purifier and the Pool of Deterministic Purifiers in Section 4.3 is brilliant.
- The experimental results show improvements compared to the competing algorithm

**Weaknesses:**

- Local implicit functions have already been employed to defend against adversarial attacks, and the proposed method of using local implicit functions to randomly purify patches in images lacks novelty.
- Randomness can disrupt gradient estimation and, as a result, interfere with gradient-based attacks, which is expected and didn't provide me with significant insights.

**Questions:**

- In the second paragraph of Section 1, should "extremely low" be changed to "extremely high"?
- In Section 4.3, "the robustness of the system is averaged across different purifiers" implies what? It lacks explanation.

---

> ### Author Response · Authors · 2023-11-19
>
> We thank reviewer bHtq for the comprehensive and detailed evaluation of our work. We appreciate the time and effort you put into reviewing our work. Please find below our responses to your comments.
>
> ## Addressing Novelty Concerns
> Our work **theoretically** and **experimentally** addresses the limitations of existing **purification** methods for **black-box defense**. We have listed the key points below:
> * Our work provides a theoretical analysis for **purification-based defense** in black-box setting. The analysis **quantifies** the relationship between the **number of purifiers** and the **robustness** of the system. (**Theoretical Contribution)**
> * Random-patch wise mechanism has **constant** inference cost with the number of purifiers and uses the most efficient purifier, which makes our mechanism the most efficient purification method. This method is suitable for real-world MLaaS systems (**million-fold inference cost**). (**Technical Contribution**)
>
> ## Elucidating Insights from the Paper
> We wish to underscore the insights from the **convergence of black-box defense** and the **protection over specific adversarial points**, which apply universally to all **purification-based methods** (preprocessor stage) for black-box defense.
> * Our work proves that **a single deterministic purifier** cannot improve the system's robustness against black-box attacks from the above two aspects.
> * To address this issue, we design a random patch-wise method to **maintain a constant inference cost** regardless of the number of purifiers.
> * We hope the insights from our theoretical analysis will be instrumental in **designing purification-based methods** for black-box systems, which is now emphasized in our contribution section (Section 1) for further clarity in the revised version of the paper.
>
> ## Response to Questions
> We apologize for any confusion caused by the presentation of our paper and appreciate your patience.
>
> ### Question 1: "extremely low" and "extremely high"
> The term 'extremely low' in Section 1's second paragraph refers to the **impracticality** of deploying **resource-intensive** defense algorithms against query-based attacks in **real-world** scenarios. Following this sentence, **'a million-fold cost'** for large corporations is used to illustrate the impracticality of the defense mechanism.
>
> ### Question 2: "the robustness of the system is averaged across different purifiers"
> We understand the ambiguity of this phrase and have revised the sentence to clarify its meaning:
>
> "While adversarial optimal points persist, the presence of multiple optimal points under different purifiers serve as the first trial to enhance the robustness of all purification-based methods."
>
> We acknowledge the challenge of identifying adversarial points for any purification function and present our solution as a pioneering effort to obscure these points, thereby bolstering the system's robustness.

---

> > ### Comment · Reviewer_bHtq · 2023-11-22
> > **Thanks for the response**
> >
> > Thank you for the response. Most of my questions have been addressed. I also read other reviewers' comments and the corresponding responses. I would like to discuss with other reviewers further to make the final recommendation.

---

> ### Author Response · Authors · 2023-11-21
>
> Thank you again for your time and effort in reviewing our work.
>
> There are only less than 2 days left to the rebuttal deadline, and we sincerely want to know whether our responses can successfully address all your concerns.
>
> Please also let us know if you have further concerns. We are glad to continually improve our work to address them.

---

### Official Review · Reviewer_XS3B · 2023-11-01

**Soundness:** 3 good
**Presentation:** 2 fair
**Contribution:** 2 fair
**Rating:** 5
**Confidence:** 4

**Summary:**

The paper proposes to defend black-box attacks by input purification. The key design is a random patch-wise strategy with an ensemble of lightweight purification models. The authors theoretically explain the effectiveness of the method. Results on CIFAR-10 and ImageNet show the usefulness of the method.

**Strengths:**

1.	It is the first work, to my knowledge, to induce purification for black-box defense
2.	The paper makes some modifications to purification compared to existing methods
3.	The organization is good and it gives a good survey of this task

**Weaknesses:**

1. I cannot see a clear motivation from the design to the goal. The existing purification defense is for white-box attacks, and to do black-box attacks, why do the authors design a random patch-wise image purification mechanism using the local implicit function? The method seems to be a lot of improvements in purification and the design enables theoretical analysis of query-based attacks. But the technical contribution does not directly serve the goal: you can also use it to increase the robustness against white-box attacks. In 4.1, the motivation is claimed to be an ensemble multiple purifier & adversarial examples are in low-D manifold, which is about the general defense for all attacks, not motivated for query-based attacks.

2. The method also hurts accuracy. It would be helpful to study the trade-off between accuracy and defense performance in (5)

3. The presentation of the method details is not very clear.

**Questions:**

Response to rebuttal: Thanks for providing a strong rebuttal with good modifications. The motivation is clear now: For a defense without hurting accuracy, purification is induced with theoretical analysis that multiple purifiers are needed. To address the efficiency issue of ensembling multiple purifiers, the proposed method is presented. But my concerns remain because the method does NOT hurt the minimum accuracy as claimed, especially for low-res images when the purification blurs the input. According to the new results, 86% accuracy for CIFAR-10 is obtained, which is lower than adversarial training. The reported "random noise input" clean accuracy (78%) is also significantly lower than the original paper (93.6%). Therefore, I keep my score.

---

> ### Author Response · Authors · 2023-11-19
>
> We thank reviewer XS3B for the comprehensive and detailed evaluation of our work. We appreciate the time and effort you put into reviewing our work. Please find below our responses to your comments.
>
> ## Motivation for the Design
> We propose our random patch-wise purification method to facilitate the application (**low extra inference cost required**) of **multiple purifiers** (proved necessary to defend black-box attacks) in a real-world MLaaS system. We have updated our motivation in Section 4.1 accordingly and listed the key points below:
> * Our theoretical analysis has proved that **a single deterministic purifier** cannot improve the system's robustness against black-box attacks.
> * Since directly ensembling multiple purifiers will **linearly increase** the inference cost, we propose a random patch-wise method to **maintain a constant inference cost** regardless of the number of purifiers. It is the key point since extra inference cost for each image lead to a **million-fold** cost increase in real-world MLaaS systems.
> * Our method is the purification method with the **lowest inference cost** among all the purification methods as illustrated in Table 1 in the paper.
> * We can surely use this method to defend **white-box attack**, which is not the focus of our paper.
>
> ## Trade-off between accuracy and defense performance.
> We have conducted detailed experiments to determine the clean accuracy of using **each** purification model available and the **whole** defense mechanism, which can be found in Appendix I of the revised paper. The conclusion is that, either the usage of single purification model or the entire defense mechanism will have barely **minimum impact** on the **clean accuracy** of the classifier.
>
> ## Presentation of the method detail.
> We have revised our paper for presentation of the method detail and found that the **clarity** of our paper is influenced by **lack** of explanation and annotations on figures as suggested by reviewer cfuj. In the revised paper, we have made the following changes to improve the clarity of our paper:
> * **Figure 2:** We have circled the main body of a purifier model for later reference.
> * **Figure 3:** We have added annotations to the figure to better illustrate the difference of our method and the ensemble method.
> * **Figure 4:** We have added more details to present the training process of our method. The caption is also revised for better illustration.

---

> ### Author Response · Authors · 2023-11-21
>
> Thank you again for your time and effort in reviewing our work.
>
> There are only less than 2 days left to the rebuttal deadline, and we sincerely want to know whether our responses can successfully address all your concerns.
>
> Please also let us know if you have further concerns. We are glad to continually improve our work to address them.

---

### Meta-Review · Area_Chair_bpvh · 2023-12-09

**Metareview:**

This work studied the defense against the query based adversarial attacks. It proposed a method by utilizing the adversarial purification, as well as the ensemble  mechanism. It was claimed efficient and effective.

There were 4 detailed and informative reviews, and there were several discussions between authors and reviewers.

The contributions of utilizing adversarial purification for black-box defense, and the theoretical analysis (but similar with the analysis in existing black-box defense) are recognized by some viewers. However, there are several important concerns, as follows:
1. The experimental setting and reported results are questionable.
2. The experimental results are not satisfied. As shown in Table 2, the proposed method didn’t show superiority to existing works.
3. The complexity is much higher than several existing methods.


In summary, the performance (both effectiveness and efficiency) of the proposed method is not strong, and its value to the community is questionable.

**Justification For Why Not Higher Score:**

The performance (both effectiveness and efficiency) of the proposed method is not strong, and its value to the community is questionable.

**Justification For Why Not Lower Score:**

n/a

---

### Decision · Program_Chairs · 2024-01-16

Reject